# Subthreshold Nano-Second Laser Treatment and Age-Related Macular Degeneration

**DOI:** 10.3390/jcm10030484

**Published:** 2021-01-28

**Authors:** Amy C. Cohn, Zhichao Wu, Andrew I. Jobling, Erica L. Fletcher, Robyn H. Guymer

**Affiliations:** 1Centre for Eye Research Australia, The Royal Victorian Eye and Ear Hospital, Melbourne 3002, Australia; wu.z@unimelb.edu.au (Z.W.); rh.guymer@unimelb.edu.au (R.H.G.); 2Department of Ophthalmology, Department of Surgery, The University of Melbourne, Parkville 3052, Australia; 3Department of Anatomy and Neuroscience, The University of Melbourne, Parkville 3052, Australia; aij@unimelb.edu.au (A.I.J.); e.fletcher@unimelb.edu.au (E.L.F.)

**Keywords:** drusen, age-related macular degeneration, laser

## Abstract

The presence of drusen is an important hallmark of age-related macular degeneration (AMD). Laser-induced regression of drusen, first observed over four decades ago, has led to much interest in the potential role of lasers in slowing the progression of the disease. In this article, we summarise the key insights from pre-clinical studies into the possible mechanisms of action of various laser interventions that result in beneficial changes in the retinal pigment epithelium/Bruch’s membrane/choriocapillaris interface. Key learnings from clinical trials of laser treatment in AMD are also summarised, concentrating on the evolution of laser technology towards short pulse, non-thermal delivery such as the nanosecond laser. The evolution in our understanding of AMD, through advances in multimodal imaging and functional testing, as well as ongoing investigation of key pathological mechanisms, have all helped to set the scene for further well-conducted randomised trials to further explore potential utility of the nanosecond and other subthreshold short pulse lasers in AMD.

## 1. Introduction

Tremendous advances have been made in the treatment of neovascular age-related macular degeneration (AMD) with the introduction of anti-vascular endothelial growth factor (anti-VEGF) intravitreal injections [1,2,3]. However, there has been very little advance in our ability to intervene in the early or intermediate stages of the disease, in order to prevent or slow disease progression. There remains an urgent unmet need for proven, efficacious intervention strategies at earlier stages of AMD to prevent progression to vision-threatening, late stages of this common and devastating disease [4].

Drusen are extracellular, lipid-rich deposits that accumulate over time in between the retinal pigment epithelium (RPE) and Bruch’s membrane (BM) and are one of the earliest clinical hallmarks of AMD, representing an important biomarker for risk of disease progression to vision threatening late complications of AMD [5]. Drusen composition using histological markers has been well documented. Although drusen are known to contain carbohydrates [6], zinc [7], proteins [8,9,10] and constituents of the complement system [11], the largest component is lipids [12,13,14,15]. Another well-known hallmark of AMD—albeit one not readily imaged in the clinic, but seen histopathologically—is thickening of the BM, where an accumulation of lipid-rich debris reduces essential transport across the membrane [16,17,18,19]. With advances in multi-modal imaging, in particular optical coherence tomography (OCT), other biomarkers have been identified that confer an increased risk of AMD disease progression, including reticular pseudodrusen (RPD) [20,21,22,23,24], hyper-reflective foci [25], drusen with heterogeneous internal reflectivity [26] and nascent geographic atrophy [27]. These features have enhanced our understanding of the disease stage, the risk of progression, and the appreciation of various clinical phenotypes within AMD. This is especially evident with the increasing appreciation of RPD—both in its prevalence and potential underlying pathophysiology [23,24]. This new, more granular ability to phenotype the disease, will likely need to be considered as we work towards targeted intervention strategies to prevent progression to late atrophic or neovascular AMD complications.

The time of progression from the development of drusen to vision-threatening late stage complications is often many decades, providing a large window of time in which to intervene to slow progression. Laser, in particular its non-thermal application through subthreshold, very short pulses, offers a potential therapeutic option to explore. In this review, we discuss the evolution of laser use in AMD from the early observations using continuous wave (CW) thermal lasers through to the newer short pulse, subthreshold laser treatment trials and histological findings. We also present a body of preclinical work that explores the potential mechanism of action of a nanosecond laser that provides a rational for its possible efficacy in slowing AMD disease progression.

## 2. Historical Use of Ophthalmic Lasers in Age-Related Macular Degeneration

As early as the 1970s, Gass made the incidental observation that when thermal, continuous wave (CW) ruby or argon lasers were applied to the retina for diabetes, drusen were noted to regress [28]. As a result of these observations, several trials in the 1990s sought to explore whether thermal CW laser treatment could be used to slow AMD disease progression, with the hypothesis being that clearance of drusen could alter the underlying pathology of AMD, thereby slowing the progression to end-stage disease [29]. Some studies indicated a beneficial effect of lasers in preventing visual loss [30,31], while others suggested a possible increase in neovascular AMD (nAMD) [32,33]. However, a Cochrane systemic review of 11 clinical laser trials found that whilst drusen did indeed regress with laser therapy (with an odds ratio of >9), there was neither a beneficial effect in slowing AMD progression, nor evidence of an increased risk of nAMD, geographic atrophy (GA), or vision loss [34].

Hypotheses were put forward at the time to explain a potential therapeutic benefit of laser treatment in AMD. One histological report suggested an increase in, or activation of, normal choroidal endothelial protrusions which increased their surface area within Bruch’s membrane and may have resulted in greater clearance of debris [35]. Others suggested a possible release of immune mediators from the retinal pigment epithelium (RPE) [36] or induction of phagocytic cell activity [37]. However, it was also evident from natural history studies that drusen regression also occurred as part of the characteristic progression to atrophy [27,38,39], even in the absence of laser therapy. As such, witnessing the reduction in drusen alone is not sufficient to determine whether the progression of the disease has been slowed.

The application of thermal CW laser treatment may result in biological processes that influence AMD progression, but whilst standard CW laser energy is absorbed by melanosomes within the RPE, it is also converted to thermal energy within the RPE and choroid, resulting in collateral or bystander damage in the adjacent neuro-retina [40,41]. This destruction is thought to give conventional thermal CW laser its therapeutic effect (known as “laser induced retinal damage”, or LIRD) for a wide variety of diseases such as retinal ischaemia [42,43,44], diabetic macula oedema [45], polyps in polypoidal choroidal neovascular membranes [46], and retinal artery macro-aneurysms [47], where the aim is often to destroy tissue. However, this type of collateral damage is disadvantageous when the aim of treatment is to maintain the structure and function of surrounding cells, such as the photoreceptors, BM, and other neural cells. Thermal CW laser photocoagulation also results in an overt inflammatory response within the retina, resulting in exacerbation of retinal damage [48]. In particular, thermal injury has been reported to upregulate inflammatory mediators such as cytokines [49,50] and growth factors, [36,51] while retinal glia exhibit an increase in gliotic markers, including the intermediate filament and glial fibrillary associated protein (GFAP) [51,52].

Given the lack of any evidence for a beneficial effect of thermal CW lasers in reducing progression to late-stage AMD and the collateral retinal damage that ensues, potentially resulting in scotomas and increased risk of choroidal neovascularization (CNV) [40,53], these lasers were not pursued as a treatment option for early and intermediate AMD. The damaging effects of thermal CW lasers restricts the majority of their use to the peripheral retina, where secondary off-target cell damage is of less significant clinical consequence.

## 3. Development of Newer Retinal Laser Technology to Allow Shorter Duration Pulses and a More Targeted Effect

In order to treat diseases of the macula, researchers have sought a means whereby they could harness the potential positive effects of thermal CW lasers in a way that avoided the bystander thermal damage to the neurosensory retina and choroid. The ability to restrict laser-induced effects to just the RPE was introduced by Anderson and Parrish in 1983 in a method termed “selective photothermolysis” [54]. They proposed the application of extremely brief laser pulses to the RPE to limit heat dissipation into the surrounding tissues. This led to the development of lasers with pulse durations in the microsecond range, such as the retinal laser described by Pankratov [55] that delivered laser energy in short pulses (“micro-pulse”) rather than as a continuous wave. The technology allowed for greater control over laser treatments due to the innate concept of alternating an active “on” cycle with an “off” cycle, where the duty cycle refers to the “pulsing” and is defined as the length of time the power is “on” divided by the total time the laser is used. Using this definition, a CW laser has a duty cycle of 100%, whereas a 5% duty cycle laser refers to a laser that is pulsed “on” for 100 milliseconds (ms), with a 1900 ms “off” time. The advantage of pulsed lasers is that the temperature rise within the tissue during the “on” time is dissipated during the “off” cycle [56]. Modelling has shown that the ideal duty cycle is less than 5% to maximise efficacy and safety [56,57].

The diode micro-pulse (SDM) lasers, developed in the 1990s, employed the rapid application of a burst of laser pulses with a pulse duration of 100–300 microseconds over a 100–500 ms time window. More recently, selective retinal therapy (SRT) is an approach that utilises the application of a burst of very short laser pulses of 1.4 ms in duration, and a duration between pulses of about 10 ms. Although the laser pulses in both SDM and SRT systems induce a temperature rise within the RPE (i.e., cause thermal effects), the time between each pulse is sufficient for the temperature to theoretically return to baseline, thereby reducing the potential for diffusion of heat into surrounding tissues, such as the neural retina. The thermal relaxation time, a measure of ability of thermal energy to diffuse through the cell, is calculated to be approximately 10 ms for the RPE. This suggests that intervals between pulses that are >10 ms would result in very little, if any, thermal energy diffusion into photoreceptors [58]. Thus, the length of the interval between laser pulses, together with the pulse duration, determines whether thermal damage extends beyond the RPE [59].

The mechanism(s) of cell destruction induced by short-pulsed lasers are distinct to those induced by thermal CW lasers [60]. Laser energy is absorbed by melanosomes within the RPE, and when laser pulse durations are >4 ms, there is liberation of heat within the cell that can extend into the surrounding neural retina [60]. When RPE cells are irradiated with pulse durations that are less than 4 ms, mechanical disruption of the cell is thought to occur, because heating of melanosomes is below the temperature to cause thermal effects within the cell, such as the coagulation of proteins. Rather, small bubbles of steam develop around the melanosomes within the RPE which lead to the transient expansion of the cell and ultimately mechanical disruption [60]. Based on this information, it is likely that even some short pulse lasers could induce thermal damage to surrounding tissue, whereas nanosecond and microsecond lasers could potentially deliver more selective loss of the RPE. Indeed, evidence to suggest thermal changes can be seen when using micro-pulse lasers comes from an evaluation of the heat shock proteins in the RPE, especially HSP70, an indicator of thermal changes, in response to the SDM laser [61]. More research is needed to determine the extent of any more widespread thermal effects when using pulses in the microsecond range, and what the effect of repetitive laser bursts could be if there is a gradual increase in cell temperature over time. These would be an important consideration in the application of these lasers for the treatment of macula diseases.

A laser in the range of nanoseconds has been developed (2RT^®^, Ellex Pty Ltd. Adelaide, Australia), which uses a Q-switched frequency doubled laser to deliver 3 nanosecond (ns) pulses to the posterior eye [62]. The energy absorbed by the RPE in response to these short pulses is 1/500th of that delivered by thermal CW lasers, and it employs a speckled beam, resulting in sporadic and selective loss of RPE cells [62,63]. In view of the extremely short pulse duration, the nanosecond laser provides a mechanism for inducing selective changes in the RPE in the absence of thermal cellular changes with a wide safety margin.

The precise mechanism by which lasers induce protective effects on the posterior eye remain to be definitively elucidated, but one possibility is via the release of various protective factors from the RPE. In this section, we provide an overview of the cellular effects of nanosecond laser application (2RT^®^, Ellex Pty Ltd. Adelaide, Australia) to the posterior eye. The positive effects of this laser provide the foundation for understanding how nanosecond lasers might be efficacious when used to treat macular diseases, including AMD.

## 4. Mechanisms of Action of Nanosecond Laser Treatment and Evidence of Safety in Animal Models

Although there are no naturally occurring animal models of AMD, the effects of nanosecond laser irradiation on the posterior eye and its selectivity for the RPE have been demonstrated in in vitro porcine explants, in vivo rodents, as well as in vitro in human RPE cell cultures and also in two exenterated human eyes [62,63,64,65,66]. Using porcine cultures, the level of laser energy that can be delivered before damage to the overlying photoreceptors occurs (called the therapeutic range ratio) is almost three time higher for the nanosecond laser (e.g., 3.6:1) than a thermal CW laser (1.3:1), suggesting that there is a greater safety margin with this class of laser than for thermal CW lasers. This is consistent with the notion that the nanosecond laser delivers high levels of laser energy to the RPE, but that cellular effects on neighbouring tissues is minimal in contradistinction to thermal CW lasers [65].

In all model systems evaluated to date, the nanosecond laser has been shown to selectively ablate small areas of the RPE, leaving the adjacent neuroretina intact. In the mouse posterior eye, small regions of the RPE showed restricted cell death within 5 h of laser irradiation and healing over a 1–7-day period that was characterised by an increase in individual RPE cell size within laser treated areas, as well as labelling of RPE nuclei with the proliferation marker, cyclin D1 [63]. Similarly, five days following nanosecond laser treatment, exenterated human eyes showed enlargement and migration of RPE cells into the treated area, while in vitro human RPE cells show increased labelling of the S-phase marker, bromodeoxyuridine, suggesting cell proliferation [63]. However, it should be noted that despite the indication of RPE proliferation, the formation of daughter cells in the human RPE is yet to be demonstrated. Indeed, RPE cells in laser-treated regions often show high numbers of nuclei within individual cells, suggesting that there may be a modification of nuclei numbers, but not the generation of daughter cells.

As noted above, a unique feature of the nanosecond laser is the selectivity of its effect to the RPE. This has been confirmed in in vivo rat and mouse experiments, where retinal integrity was assessed at low and high energy doses of nanosecond lasers. Using a “low” or clinically relevant nanosecond laser energy dose (0.21 mJ in rat and 0.065 mJ in mouse), minimal cell death within the outer nuclear layer was observed [64,66,67] and only a small increase in gliosis markers was observed, including increased expression of glial fibrillary acidic protein in Müller cells and increased expression of the intermediate filament, nestin [64,65]. In addition, repeat laser application in the same region of the posterior eye after three weeks did not exacerbate these retinal changes [68].

The effect of clinically relevant and suprathreshold energy doses of nanosecond laser have also been assessed in the human eye five days after laser application [63]. Nanosecond laser pulses with a clinically relevant dose of 0.3 mJ showed no disruption of the outer retina, nor localised gliosis, cell death, or activation of resident immune cells, microglia. In contrast, the application of a thermal CW laser was associated with significant disruption of the outer retina, combined with activation of innate immune cells within the subretinal space [63]. This is shown in Figure 1, where cross sections of retinas are shown corresponding to regions away from laser-treated areas, and areas treated with either a nanosecond laser or continuous wave laser. Importantly, photoreceptor disruption, which is evident in areas of CW laser treatment, is not seen in areas receiving nanosecond laser treatment.

An important consideration in the assessment of the safety of the nanosecond laser is its effect on the BM and any potential changes that might be considered to increase the risk of choroidal neovascularization. Breaches of the BM, especially the elastic lamina, could potentially increase the risk of neovascular complications [69]. Using electron microscopy to assess BM integrity in the posterior mouse eye seven days after nanosecond laser application, all five layers of Bruch’s membrane were observed to remain intact within laser-treated areas. This suggests that although RPE cells are ablated by the nanosecond laser, the effects are restricted to the RPE and do not alter the integrity of the BM [67]. In addition, the application of nanosecond pulses with a supra-threshold energy dose had no effect on the expression of the three VEGF isoforms in the mouse RPE or retina [63]. Moreover, the expression of the anti-angiogenic factor, platelet epithelium-derived factor (PEDF) was increased in both the mouse RPE and retina seven days after application of the suprathreshold nanosecond laser treatment [63].

Overall, these results confirm the safety profile of the nanosecond laser when applied to the rodent or human posterior eye. Treatment of the posterior eye with the nanosecond laser induces selective loss of the RPE, in the absence of bystander effects in the overlying neural retina or any deleterious effect on the BM. Furthermore, suprathreshold dosages do not alter signals that could potentiate choroidal neovascularization. These findings are important considering the potential for nanosecond lasers in the management of diseases of the macula, such as AMD.

## 5. Nanosecond Laser Treatment Abrogates Changes in the Posterior Eye Important in the Development of AMD

Having established that the nanosecond laser selectively ablates the RPE in the absence of damage to neighbouring structures, it is important to address its effect on the posterior eye that has the potential to reduce the progression of AMD. The formation of drusen and a thickening in the BM are critical in the development of early AMD. Investigation of BM thickness in an animal model with features of early AMD demonstrated a thinning of the BM in response to nanosecond laser application [63]. ApoEnull mice, which have a thickened BM, were treated with the 2RT^®^ laser. Ten spots were delivered in each eye at nine months of age, and eyes were then evaluated three months later. In contrast to ApoEnull mice eyes that were sham-treated and showed a substantially thickened BM (~900 nm thick), animals that had received nanosecond laser treatment to one eye showed a significant reduction in thickness (~700 nm thick) in the treated eye (Figure 2) [63].

In order to investigate the mechanism of this apparent nanosecond laser effect, it is important to realise that the BM is a dynamic structure consisting of extracellular matrix, including alternating layers of collagen and elastin. Its turnover is controlled by signalling pathways within the RPE, including the expression of matrix metalloproteinases (MMP) and tissue inhibitors of matrix metalloproteinases (TIMPs), which are important for the formation and degradation of constituents of the BM. In vitro studies on cultured human RPE cells have revealed that treatment with the nanosecond laser showed induced expression of MMP2 and MMP9, with these enzymes being released within two days of subthreshold nanosecond laser (SNL) treatment [66]. Expressional analysis of genes associated with the formation and degradation of the extracellular matrix has also been carefully examined in a mouse model with features of AMD. Changes in gene expression of 84 genes associated with extracellular matrix turnover have been examined in 12-month-old C57Bl6 (control) and ApoEnull (AMD-like model) mice three months after nanosecond laser treatment. A total of nine genes were significantly dysregulated by more than two-fold, including *Mmp*2 and *Mmp*3, a finding that was also confirmed by quantitative RT-PCR [63]. These data suggest that treatment of the RPE of aged ApoEnull mice with a nanosecond laser alters the turnover of extracellular matrix components of Bruch’s membrane by altering the expression of MMPs within the RPE [63].

One of the more intriguing findings in animals treated with the nanosecond laser was the observation that changes in gene expression in the RPE occurred in both the laser treated and the untreated fellow eye. Indeed, both *Mmp*2 and *Mmp*3 were upregulated by similar amounts in both eyes, alongside seven other genes associated with extracellular matrix turnover [63]. Although the BM was not significantly thinned in untreated contralateral eyes, these results suggest that the nanosecond laser could have distant effects, the mechanisms and significance of which require further study.

Overall, these findings suggest that nanosecond laser application selectively ablates RPE cells without inducing overt visible damage in adjacent structures. Moreover, absorption of nanosecond laser energy by the RPE induces gene expressional changes that are associated with thinning of the BM, particularly involving the MMPs. These findings support the evaluation of this laser in macular conditions, including AMD.

## 6. Human Proof of Concept Study of Nanosecond Laser Treatment in AMD

The 2RT^®^ nanosecond laser was used in a human pilot study in 2012–2013 [70]. The study recruited 50 people with bilateral large drusen (>125 μm; meeting the definition of intermediate AMD) and with best corrected visual acuity (BCVA) >20/63 in both eyes. The treatment protocol was a single session in one eye of 12 laser spots, 400 μm in diameter (at the retina) and placed >500 μm from the fovea. The 12 spots were chosen because this was a similar approach taken in one of the original thermal CW laser studies [34]. The energy was titrated for each individual by establishing a “threshold”—the energy at which a visible burn was seen. The treatment energy was then turned down from threshold so as to deliver sub-threshold laser to the macula (range 0.15–0.45 mJ; average 0.24 mJ). A natural history cohort was included for comparison and consisted of 58 untreated participants with intermediate AMD (bilateral large drusen). Both groups were followed up at six-month intervals for two years [70]. Drusen load was assessed using multimodal imaging, including colour fundus photographs, OCT, and fundus autofluorescence (FAF). Eyes that reached late disease (nAMD or GA) were excluded from the analysis of drusen load grading; the development of late disease invalidates any grading on drusen load because drusen disappear when late AMD occurs. After 12 months, 40% of eyes receiving the 2RT^®^ laser had a reduction in drusen area compared with baseline [70]. This was statistically significant when comparing the treatment cohort to the natural history group, where a reduction in drusen area only occurred in 5% of eyes (*p* < 0.001) [70]. This effect was maintained over two years, with 35% of the treated eyes demonstrating reduction in drusen area compared to 11% in the natural history group (*p* < 0.01). An interesting observation was that the untreated fellow eyes in participants receiving the 2RT^®^ laser also demonstrated a reduction in drusen area at 12 months compared to the natural history cohort (*p* = 0.05), although this effect was not maintained into the second year of follow-up [70]. Importantly, this pilot study demonstrated that laser-induced drusen resolution did not result in progression to atrophy to the two-year follow-up time point. Moreover, FAF imaging of specific regions where drusen regressed were not hypoautofluorescent, a potential indicator of progression towards GA [70]. This is important because spontaneous resolution of drusen often heralds the development of atrophy [39]. This pilot study concluded that at two years, there were resolutions of drusen in 2RT^®^-treated eyes without evidence of progression to atrophy.

## 7. Nanosecond Laser Treatment in Early Age-Related Macular Degeneration: The Laser Intervention in the Early Stages of Age-Related Macular Degeneration Study

Following the pilot study, a larger randomised clinical trial was conducted: The Laser Intervention in the Early Stages of Age-Related Macular Degeneration (LEAD) study [71,72]. The LEAD study was a 36-month, multicentre, randomised, sham-controlled trial, designed to evaluate the effect of the 2RT^®^ in individuals with bilateral large drusen. The treatment was referred to as “subthreshold nanosecond laser” (SNL). A total of 292 participants with BCVA >20/40 in both eyes were randomised to receive either SNL therapy or sham treatment to a study eye every six months and were followed over a three-year study period. The main outcome measure was the time to the development of late AMD in the study eye, as defined by MMI, which for this study was defined as colour fundus photography, OCT, FAF, and fluorescein or indocyanine green angiography (as clinically indicated) [71,72]. The LEAD trial was the first trial to use a combined atrophic endpoint of atrophy, as defined either on OCT as nascent geographic atrophy (nGA) through to GA as defined on colour fundus photography [71]. SNL treatment was applied using the 2RT^®^ laser with 12 spots, 400 μm in diameter applied to the macula area—six spots just inside the superior arcade and six spots inside the inferior arcade. Test spots were used to determine threshold energy for each participant, and then reduced to perform the treatment to ensure subthreshold delivery of the laser [71]. Sham laser was performed in the same way, but short bursts of light from the 2RT^®^ illumination system were used to simulate the laser.

Analysis of the LEAD study showed that in participants with bilateral large drusen, there was no significant difference in the overall progression to late AMD when comparing the group randomised to SNL to the group receiving sham treatment. At 36 months of follow-up, 45 patients (15.4%) developed late AMD in the study eye, with this occurring in 20 (13.6%) participants in the SNL group and 25 (17.2%) participants in the sham group. However, a proportion of individuals with bilateral large drusen also exhibited the high-risk RPD phenotype. Given that these individuals might have a more dysfunctional RPE and may not respond as well to laser treatment compared to those with conventional drusen, a post-hoc analysis was performed. The post-hoc analyses revealed evidence that the effect of the SNL treatment was modified based on the coexistence of RPD at baseline (interaction *p* = 0.002). Specifically, for the 222 (76.0%) participants without coexisting RPD at baseline, the rate of progression to late AMD showed a more than four-fold reduction in the SNL group compared to the sham group (*p* = 0.002). Conversely, in the 70 (24.0%) participants with coexisting RPD at baseline, there was potentially an increased rate of progression to late AMD in the SNL-treated arm compared to the sham arm (*p* = 0.112) [72]. This was the first study to suggest a potential differential result of an intervention based upon the AMD phenotype with regards to the drusen subtype. Whilst these findings from the post-hoc analysis should be interpreted with caution, they are biologically plausible and provide an hypothesis that this form of laser treatment may be effective at slowing disease progression in those with intermediate AMD without RPD. However, this hypothesis requires validation in future, larger randomised trials [72].

## 8. Secondary Outcomes and Other Research from the LEAD Study

The secondary and exploratory outcomes of the LEAD study looked at the time to develop late AMD in the non-study eye based on MMI, change in visual function (BCVA, LLVA and microperimetric sensitivity) and drusen volume in the study and non-study eyes, and participant-reported outcomes from the Night Vision Questionnaire (NVQ-10) and Impact of Vision Impairment (IVI) questionnaire [73]. Overall, SNL treatment did not significantly delay overall progression to late AMD in the fellow non-study (untreated) eye. Although not significant, there was a trend of effect modification based on the coexistence of RPD in the non-study eye (interaction *p* = 0.065), with similar trends as seen in the study eye (i.e., reduced rate of progression for those without coexistent RPD with SNL treatment, while a potentially increased rate of progression in those with RPD). These findings are consistent with the preclinical animal studies that revealed a thinning of the BM in the lasered eye and changes in gene expression in both the lasered eye and non-lasered fellow eye, suggesting a possible systemic effect.

There was no significant difference in the change in the visual function measures and drusen volume in both the study and non-study eyes, and no significant difference in the participant-reported outcomes, between those in the SNL and sham treatment groups. The only exception was a slightly greater drop in BCVA in the SNL group compared with the sham for study eyes, but no consistent reasons for this finding were found based on a clinical review of the cases showing a ≥10-letter drop. Furthermore, this observation did not correlate with other visual function measures, but indeed warrants further evaluation [73]. The absence of evidence for a reduction in drusen load for study eyes in the SNL treatment group was unexpected [73] as this had been observed in our pilot study [70]. However, these findings suggest that the potential positive effects of SNL treatment on delaying progression to late AMD may not necessarily be reflected by changes in drusen load. This adds complementary knowledge to the previous findings that, despite inducing drusen regression, CW thermal lasers did not lead to beneficial effects for slowing late AMD development [34], highlighting the importance of evaluating late AMD development as the main outcome in such trials.

Further examination of the potential impact of SNL treatment parameters on the progression to late AMD was performed using the data from the LEAD study. The lack of real-time visual feedback of a sub-threshold laser application can make it difficult to determine if the treatment has been delivered adequately to the RPE to bring about the desired effect. Whilst the laser spots were not clinically visible at the time of the intervention, they were often readily visible on FAF imaging obtained at subsequent visits in the LEAD study. We therefore sought to examine if there was a “dose–response” relationship between the number of visible spots on FAF imaging, as well as the treatment laser energy used from the first two LEAD treatments on late AMD progression over the three-year study period. Multivariable analyses revealed that there were no significant associations between the time to develop late AMD and number of FAF-visible laser spots, nor the laser energy used during the SNL treatments, delivered early in the trial. Thus, there was no evidence to suggest that a dose–response relationship existed for the effect of laser treatment using the LEAD study treatment parameters on the progression of AMD (unpublished data).

## 9. Future Applications and Directions of Subthreshold Laser Treatments for Treating Macula Disease

Extensive research into short duration lasers have heralded the development of selective retinal therapy (SRT) and subthreshold diode micro-pulse (SDM) and nanosecond lasers. Although the use of many short duration lasers has been explored for use in retinal disease, to the best of our knowledge, the 2RT^®^ laser developed by Ellex (now Nova Eye Medical, Pty Ltd. Fremont, CA, USA) represents the only laser functioning in the nanosecond range for ophthalmic use. As such, the results of the LEAD study are only applicable for use with such a laser and cannot be extrapolated for other short pulse lasers. In addition, the LEAD study is, to the best of our knowledge, the only large randomised-controlled trial to examine the potential efficacy of a subthreshold, nanosecond laser in slowing progression of intermediate AMD to advanced disease. Nova Eye Pty Ltd. (Fremont, CA, USA) plans to continue its research using the 2RT^®^ laser in management of iAMD.

Another development in short duration laser use for ophthalmic conditions is the release of the R:GEN laser by Lutronic Vision (South Korea). The R:GEN is an SRT laser with a 527 nm wavelength and 1.7 µs pulse duration designed to selectively target the RPE, with its effect delivered through microbubble formation in the RPE. As discussed previously, lasers delivered at subthreshold levels have no visual feedback at the time of application, which can make the titration of laser power for adequate tissue effect extremely difficult. The R:GEN laser utilises Dual Dosimetry technology to measure reflectometry (back-scattered light) and opto-acoustic signalling (thermo-elastic pressure waves) to offer real-time titration of laser energy delivery to the RPE. Opto-acoustic (OA) imaging technology (also known as photo-acoustic imaging) is a non-invasive way to determine the temperature rise in the RPE cell at the time of laser treatment, utilising both light and sound wave principles. When short duration laser light is absorbed by chromophores within a tissue (such as melanosomes within the RPE), the cell undergoes thermoelastic expansion and acoustic waves are generated. These optoacoustic signals can be measured by an ultrasonic transducer. During irradiation of the RPE, the baseline temperature of the cell increases, resulting in a change to the pressure signal and acoustic waves emitted and microbubble formation can be detected within the RPE cells using OA techniques [74,75]. These methods can then indicate when sufficient energy has been generated by the laser within the RPE cell, and at this point the laser automatically switches off. It is possible that this will result in a more accurate, individualised titration of laser energy delivery. The R:GEN laser has already been studied in macular disease central serous chorioretinopathy with promising results [76], and further studies are planned in other diseases.

Another difficulty in conducting interventional trials for the early stages of AMD is the natural history of the disease itself. The disease progresses slowly over years, which renders reaching clinically meaningful results within a reasonable time frame difficult. Significant advances have been made to address this, through describing potential early disease endpoints. Nascent geographic atrophy (nGA) is one such early biomarker, signifying early atrophic changes as seen on OCT imaging. These changes were incorporated into a combined atrophic endpoint in the LEAD study (the first trial to do so), and in so doing, enabled a more time- and cost-efficient study to be conducted [25]. Similarly, the Classification of Atrophy Meeting (CAM) international consensus group have proposed a classification of atrophy defined on OCT features of both incomplete and complete retinal pigment epithelium and outer retinal atrophy (iRORA and oRORA, respectively) in AMD [77,78]. Having consensus on nomenclature around early atrophic changes in AMD will help facilitate early intervention studies, making it more feasible to assess the efficacy of novel early interventions.

## 10. Conclusions

Apart from lifestyle modifications and dietary supplements, there are no specifically targeted treatments to slow the progression from the early stages of AMD to advanced disease. Whilst nanosecond laser is not yet a recognised intervention for AMD, we have reviewed a body of work to demonstrate that it may offer an intervention where there currently is none. Preclinical models show that a nanosecond laser (2RT^®^) can be safely delivered to the retina where it is selectively taken up by RPE cells, and this treatment shows biological plausibility given our current understanding of AMD pathogenesis. The LEAD study provided clinical results supporting continued research into the potential of subthreshold delivery of nanosecond laser to provide a possible early intervention to slow AMD progression. Further well-conducted, randomised clinical trials are required to determine the efficacy and safety of the 2RT^®^, as well as all other lasers aiming to target this indication.

## Figures and Tables

**Figure 1 jcm-10-00484-f001:**
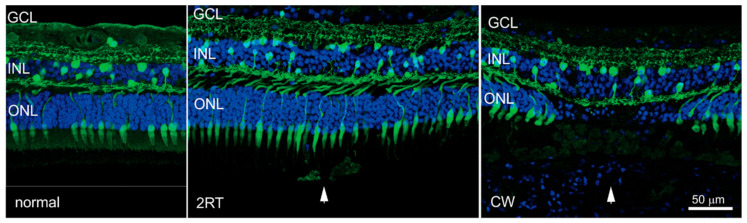
Human retinas treated with continuous wave (CW) or nanosecond laser irradiation. Cross sections of human retinas are shown immunolabelled for the neuronal marker calretinin (green), and the nuclear maker bisenzimide (blue). Sections corresponding to an area well away are shown in Figure 2. RT^®^ (2RT) or CW treatment (CW). Significant disruption of photoreceptors is evident in the region treated with the continuous wave laser. Abbreviations: ONL—outer nuclear layer; INL—inner nuclear layer; GCL—ganglion cell layer. Figure adapted from Jobling et al. (2005).

**Figure 2 jcm-10-00484-f002:**
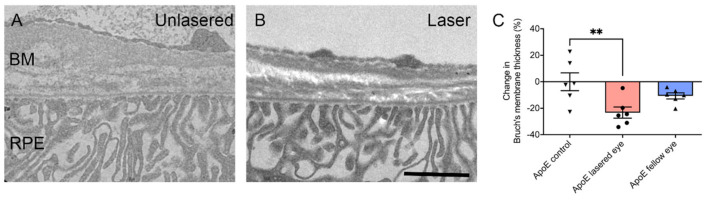
Nanosecond laser thins Bruch’s membrane in a mouse model with features of early AMD. (**A**,**B**) Electron micrographs of the Bruch’s membrane (BM) of a non-laser treated 12-month-old ApoEnull mouse and an ApoEnull mouse that had received nanosecond laser treatment 3 months prior to fixation. Abbreviations: BM-Bruch’s membrane; RPE-retinal pigmental epithelium. (**C**) Graph showing percentage change in Bruch’s membrane thickness in control, lasered, and unlasered fellow eyes of ApoENull mice that had received laser treatment 3 months prior. Application of the nanosecond laser induced significant thinning of Bruch’s membrane compared to control or fellow unlasered eyes (one-way ANOVA, Tukey’s post-hoc test; ** *p* < 0.01). Figure adapted from Jobling et al. (2005).

## Data Availability

Data sharing not applicable. No new data were created or analyzed in this study. Data sharing is not applicable to this article.

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
