# Peer review of "Subthreshold Nano-Second Laser Treatment and Age-Related Macular Degeneration"

_jcm, 2021, doi:10.3390/jcm10030484_

Round 1

Reviewer 1 Report

The authors present a comprehensive review of sub-threshold laser for the treatment of drusen. 

General comments:
1. A brief addition to the Intriduction would include a list of the principal components of drusen.

2. For the post-hoc analysis beginning line 316, would the authors consider using table rather than the narrative?

3. Again, for the Secondary measures, rather than a narrative, would the authors consider a table of the principle findings?

4. Under heading 3, would the authors consider a brief sentence or two to say that there are no naturally occurring animals models of AMD? The safety studies mentioned in 3 were performed on normal animals.

Specific comments:

1. Line 169: The two sentences beginning “However, …” are these the same references as in #55, 59 if not then please indicate or add either reference (not both likely).

2. Line 300: Please define MMI

3. Line 311-313: 20/146= 17.1%; 20/146 = 13.7 (13.6886). What is the significance of this difference? In other words, what statistical test was applied to this result?

Reviewer 2 Report

‘Subthreshold nano-second laser treatment and Age- 1 related macular degeneration.’

This is a detailed and well-researched article. The authors have done a lot of preliminary work in the field, and this is to be congratulated.

However, there is the impression that the effect of nanosecond laser therapy has already been proven. This is not at all the case and should be made clear in the article, especially in the Summary.

This does not detract from the articles scientific achievement. In the meantime, however, nanosecond laser treatments are already offered worldwide for high costs; it is not a service provided by insurance companies. It is suggested that the treatment is effective, but this cannot be said on the basis of the data. It would be fatal for the authors, but also for the manufacturer (Ellex), if one day it should turn out that nanosecond laser treatment is not effective. Despite all the euphoria and enthusiasm, I advise more cautious formulation.

Remarks:

- Is laser treatment of the RPE already too late when drusen are visible?

- Is the regeneration of RPE cells and Bruch's membrane in old humans comparable to that of animals?

- Line 293: The study cited has already been discussed [1]. But the protocol may be discussed as it would have been better to compare the treated eye with the partner eye to show intraindividual effects and exclude interindividual differences. 

Line 363: same results in fluoresceine angiography?

  1. Jobling, A.I.; Guymer, R.H.; Vessey, K.A.; Greferath, U.; Mills, S.A.; Brassington, K.H.; Luu, C.D.; Aung, K.Z.; Trogrlic, L.; Plunkett, M.; et al. Nanosecond Laser Therapy Reverses Pathologic and Molecular Changes in Age-Related Macular Degeneration without Retinal Damage. FASEB J 2015, 29, 696–710, doi:10.1096/fj.14-262444.

Round 2

Reviewer 2 Report

All changes have been done